# Patient-provider experiences with chronic non-communicable disease care during COVID-19 lockdowns in rural Uganda: A qualitative analysis

Peter K. Olds[1]*, Gabriel Nuwagaba[2], Paul S. Obwoya[2], Edwin Nuwagira[2], Jessica E. Haberer[1], Samson Okello[2,3]

1 Massachusetts General Hospital, Harvard Medical School, Boston, MA, United States of America, 2 Mbarara University of Science and Technology, Mbarara, Uganda, 3 University of North Carolina, Chapel Hill, NC, United States of America

* polds@mgh.harvard.edu

## Abstract

Non-communicable diseases (NCDs) are a growing health burden in Sub-Saharan Africa and especially Uganda, where they account for over one third of all deaths. During the COVID-19 pandemic, public health control measures such as societal "lockdowns" had a significant impact on longitudinal NCD care though no studies have looked at the lived experience around NCD care during the pandemic. Our objective was to understand the experience of NCD care for both patients and providers in southwestern Uganda during the COVID-19 pandemic. We conducted in-depth, in-person qualitative interviews with 20 patients living with hypertension, diabetes, and/or cardiac disease purposefully selected from the outpatient clinics at Mbarara Regional Referral Hospital and 11 healthcare providers from public health facilities in Mbarara, southwestern Uganda. We analyzed transcripts according to conventional content analysis. We identified four major themes that emerged from the interviews; (1) difficulty accessing medication; (2) food insecurity; (3) barriers to the delivery of NCD clinical care and (4) alternative forms of care. Pre-existing challenges with NCD care were exacerbated during COVID-19 lockdown periods and care was severely disrupted, leading to worsened patient health and even death. The barriers to care were exacerbations of underlying systemic problems with NCD care delivery that require targeted interventions. Future work should leverage digital health interventions, de-centralizing NCD care, improving follow-up, providing social supports to NCD patients, and rectifying supply chain issues.

## Introduction

Non-communicable diseases (NCDs) are a growing health burden in sub-Saharan Africa (SSA) with deaths from NCDs projected to outpace infectious causes by 2030 [1, 2]. While NCDs account for over one-third of disability-adjusted life years (DALYs) in low-income

**Data Availability Statement:** All relevant data are within the paper and its Supporting Information files.

**Funding:** This work was funded by the Wyss Global Health Fellowship awarded to PO from the Massachusetts General Hospital Center for Global Health. https://globalhealth.massgeneral.org. The funder played no role in study design, data collection and analysis, decision to publish, or preparation of the manuscript.

**Competing interests:** JEH reports consulting fees from Merck and stock ownership in Natera. All other authors declare none. This does not alter our adherence to PLOS ONE policies on sharing data and materials.

countries and represent a significant burden of hospital admissions, only an estimated 1.5% of development assistance in 2015 was allocated to combat NCDs [3, 4]. In Uganda, 33% of deaths were attributed to NCDs in 2016 and most facilities are currently insufficiently equipped to effectively manage NCDs [5, 6].

The COVID-19 pandemic has had a significant impact on healthcare systems across the world. In response to the pandemic, countries took a variety of approaches to mitigate COVID-19 viral spread, including socio-geographical "lockdowns" where transportation and social services were severely limited, and businesses and institutions were temporarily closed [7]. Due to weaker health systems, SSA was especially vulnerable to severe disruptions in healthcare services from lockdowns with about 80% of African countries reporting disruption in at least one essential health service between May and September 2021 [8, 9]. Prior work has shown significant disruptions in critical health services for malaria, human immunodeficiency virus (HIV), and tuberculosis (TB) [10–12].

For patients with chronic NCDs, disruptions during lockdowns have a high risk of leading to reduced productivity, clinical deterioration, and increased morbidity and mortality. Given their reliance on chronic, longitudinal care, this population is especially vulnerable to unreliable health systems and disruptions in care provision. Worsening in NCD care from the COVID-19 pandemic may prevent SSA from reaching the Sustainable Development Goal 3.4, which aims to reduce premature mortality from NCDs by one third by 2030 [13, 14].

As we look towards improving NCD care in the face of the rapidly growing burden of disease and the ongoing threat of future pandemics, we must take a holistic and patient-centered approach to improving NCD care delivery. The World Health Organization (WHO) recommends that patient-centered care be the foundation for successful and sustainable NCD interventions [15, 16]. In developing patient-centered care for patients with NCDs, healthcare providers must address well-documented challenges with reliable access to medications and clinicians, transportation to clinics, as well as a dearth of electronic medical records to facilitate longitudinal follow-up [17–19].

To help meet some of these barriers, there are calls among the medical community for Integrating mobile technology (mHealth) into NCD care. Cell phone use is rapidly growing in SSA, with 93 mobile cellular subscriptions per 100 people in 2021 [20]. Cell phones provide opportunities for linking providers in clinics with patients in the community, extending opportunities for clinical intervention. Cellphone-based interventions have the potential to successfully provide appointment reminders, adherence support, counseling, and disease education, though uptake in SSA remains sparse [21].

The disruptions in healthcare services that occurred as a result of societal lockdowns during the COVID-19 pandemic represent important opportunities for assessing the true impact of longitudinal care disruption for chronic NCD patients. However, no studies to date have looked at the lived experiences of patients and providers with chronic NCDs in SSA during COVID-19 lockdowns. We conducted a qualitative assessment of patients with hypertension, diabetes, and heart failure as well as the providers who care for them in southwestern Uganda, to evaluate how barriers to care changed during the COVID-19 lockdowns and understand perspectives on how to improve longitudinal NCD care, with a focus on mHealth integration.

## Methods

### Study setting and population

We conducted a qualitative study at public funded health facilities in Mbarara, Uganda. Patients were identified from the outpatient department of Mbarara Regional Referral Hospital

(MRRH), which provides NCD care to over 3000 patients with subspecialty clinics for a variety of chronic NCDs (e.g., hypertension, diabetes, cardiology). Due to the limited reliability in access to NCD care and medications in the region, longitudinal NCD care remains centralized, with significant proportion of patients seeking care at centralized hospitals—like the MRRH outpatient clinic—where resources are more often available [19, 22].

Healthcare providers were recruited from the MRRH outpatient department, one level-four (HC IV) and three level-three (HC III) health centers within the catchment area of MRRH including Mbarara Municipal HC IV, Kakoba HC III, Nyamitanga HC III, and Kasana HC III. Of note, HC IV facilities serve a county (about 100,000 people), with outpatient care including for NCDs, maternal and child health services, inpatient care, and an operative theatre for limited, emergency surgeries (i.e. caesarian sections), whereas HC III facilities serve roughly 10,000 people, providing simple medical, diagnostic, and maternal health services [23]. Staffing at health care facilities in Uganda is inadequate, with providers working multiple roles, often outside of their specialty [24]. Thus, despite defined roles, providers at health centers often have significant knowledge of NCD care, either through task shifting and providing direct care —for midwives—or by managing other aspects of NCD care and/or participating in health center management meetings—for medical records officers and laboratory technicians. Recruitment and interviews occurred between April 5, 2022, and May 17, 2022. We used COREQ (COnsolidated criteria for REporting Qualitative research) guidelines in preparing our report [25].

## Sampling and recruitment

We purposively recruited patient participants in conjunction with the clinic staff, who used clinic registries to identify participants with a similar ratio of men and women. We aimed for a balanced distribution among participants with hypertension, diabetes, and/or heart failure with several participants with comorbid HIV. This was a convenience sample, though given the centralized nature of NCD care in southwestern Uganda we describe above, the MRRH outpatient clinics treat a relatively representative sample of patients in the region. Potential candidates were approached for written informed consent. Patient recruitment continued until thematic saturation was reached on analysis.

Healthcare provider participants were, in turn, purposively recruited in conjunction with the head of each health facility. This was a convenience sample and, since we did not interview heads of facility, snowball sampling was not utilized. We aimed for a balanced distribution of participants from rural and urban clinics. Potential candidates were again approached for written informed consent. Provider recruitment continued until thematic saturation was reached on analysis.

Patient participant inclusion criteria included an established physician-made diagnosis of hypertension, diabetes, and/or heart failure, being 18 years or older, and able to speak either English or one of the local languages (Runyankole or Rukiga). Exclusion criteria included being pregnant, as pregnancy-related disease was considered beyond the scope of the current study, as well as inability to provide informed consent, including impairment from intoxication or psychosis.

Provider participant inclusion criteria included being a registered healthcare provider in a publicly funded health facility for at least 1 year, being 18 years or older, being actively involved in and knowledgeable of the care of NCD patients, and able to speak either English or one of the local languages (Runyankole or Rukiga). Provider participants unable to provide informed consent, including impairment from intoxication or psychosis, were excluded from the study.

## Data collection

Over 3 months, we conducted one in-depth, individual qualitative interviews to explore the participants' experiences with having a chronic disease, the impacts of the lockdown on their lives and medical care, and their views on how chronic care could be improved in Uganda. Interviews were conducted in an office away from the clinic to ensure privacy and confidentiality. A trained research assistant (GN) fluent in English, Runyankole, and Rukiga conducted all interviews following a prespecified interview guide S1 File. The interviewer, GN, is bachelor trained, trilingual, and a non-clinician. The interview guide was developed by author PKO with support from PSO, EN, JEH, and SO. Authors PSO, EN, and SO are Ugandan internal medicine physicians who work at MRRH, with intimate knowledge of NCD care in the region. PKO is an American internal medicine physician who has conducted research and provided clinical care in Uganda, Madagascar, Malawi, and Rwanda since 2011. JEH is American internal medicine physicians who has conducted research in Uganda, Kenya, and South Africa since 2008. The interview guide was developed through review of previous studies and discussing key aspects of NCD care that could be influenced by COVID-19 lockdowns. We used a grounded approach to avoid biasing participant responses. The initial interview guide was tested with one informal patient interview with improvements made to increase clarity. Interviews were roughly one hour in length and audio-recorded. Interviews in Runyankole were translated and transcribed from Runyankole into English by the interviewer. Interviews in English were transcribed directly in English. All transcripts were reviewed for quality.

## Data analysis

We analyzed qualitative transcripts according to conventional content analysis [26]. Author GN reviewed the first five transcripts, analyzed content to develop labels, and then created operational definitions and developed a codebook with selected illustrative quotes. PO did not have access to information that could identify individual participants. For quality control, approximately 20% of interviews were double-coded (authors GN and PO) and any discrepancies were discussed until consensus was obtained. The codebook was then refined using an iterative process as further interviews were coded. Following completion of the codebook, the remaining transcripts were manually coded. These codes were used to create a descriptive analysis based on the themes generated from the data and the goals of the study. Quotes were selected to reflect each category and illustrate themes.

## Ethical approval

This study received approval through the Mbarara University of Science and Technology Research Ethics Committee (protocol No 28/10-20), the Uganda National Council for Science and Technology (registration number HS1535ES), and the Mass General Brigham Institutional Review Board (2021P001772).

## Results

We interviewed 20 patients, of whom 10 (50%) were women and the average age was 58 (standard deviation 14), 4 (20%) had hypertension alone, 1 (5%) had diabetes alone, 2 (10%) had heart failure alone, 7 (35%) had hypertension and diabetes, 1 (5%) had hypertension and HIV, 3 (15%) had hypertension and heart failure, and 2 (10%) had hypertension, diabetes, and HIV (Fig 1).

We interviewed 11 providers, of whom 9 were women (82%). Two (18%) were from the MRRH outpatient clinic, 2 (18%) from Mbarara Municipal HC IV, 3 (27%) from Nyamitanga

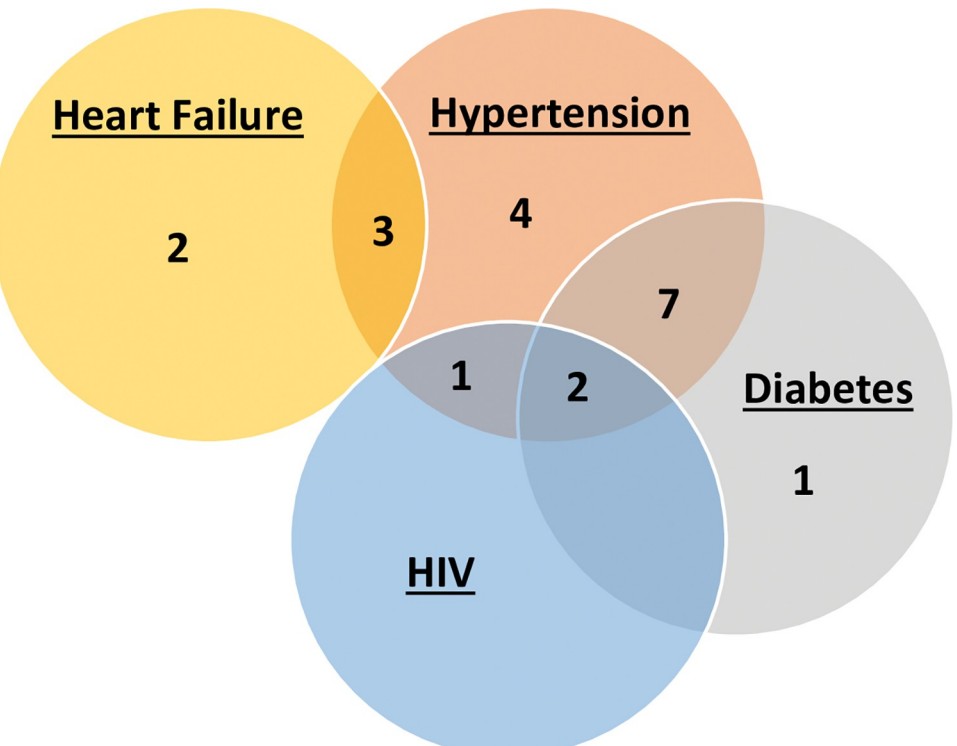

**Fig 1. Distribution of patient diagnoses.** A Venn diagram of patient diagnoses with numbers representing the total number of patients within each category. HIV: Human immunodeficiency virus.

HC III, 3 (27%) from Kakoba HC III, and 1 (9%) from Kasana HC III. Of interviewed providers, 8 (73%) were nurses, 1 (9%) was a midwife, 1 (9%) was a medical records officer, and 1 (9%) was a laboratory technician.

## Overview of qualitative interview results

We identified 4 major themes that emerged from the interviews which included (1) difficulty accessing medication, (2) food insecurity, (3) barriers to the delivery of NCD clinical care and, (4) alternative forms of care. We found that preexisting challenges with NCD care were exacerbated during COVID-19 lockdown periods and care was severely disrupted leading to worsened patient health and even death. Ultimately, patients and providers made recommendations for improving NCD care, including decentralizing care, integrating technology, and leveraging the care system developed for treating HIV.

**1. Difficulty accessing medication.** Accessing affordable medications was a major barrier to the continuum of care. Patients and providers routinely mentioned that health facilities lacked adequate medications for NCDs, which necessitated patients purchase them at often exorbitant prices from private pharmacies. However, most patients routinely mentioned that they were unable to buy medications from local pharmacies and therefore often went without medication.

> "*The only service I get from this facility is seeing and consulting health workers. Sometimes they do physical examination and health education. But anything called medicines, we buy them from the pharmacies outside the hospital.*" Patient (PA) 17

*"Hypertension drugs are so expensive, most of the time I do not buy enough dosage because I do lack money. Even the little I get access to gets finished so fast, I am forced to do without drugs for some time. By the time I come back to the hypertension clinic on my clinic day, the health workers find it [blood pressure] has already shot up." PA 5*

When discussing medication stock-outs at public facilities, providers routinely tried to find creative ways of getting care to as many patients as possible.

*"We are supposed to be giving our patients medicines for one month but because we do not have enough, we are forced to give them only for a half a month because we do want other patients also to get something. So what we do we give them fifteen tablets and tell them to go and buy the rest from outside. Then the other fifteen we give them to another patient also to survive on." Provider (PR) 6*

When discussing how care changed during the COVID-19 lockdowns, patients and providers highlighted that access to medications worsened significantly. As noted above, most patients we interviewed rely heavily on free medications from public clinics. However, the global and local supply chain disruptions during the COVID-19 pandemic worsened medication stock-outs at local health facilities.

*"The Ministry of Health would take a very long time to bring medications to the facility and that impacted us tremendously. I think there were a lot of bureaucracies securing medicine from national medical stores and that caused a lot of delay in delivering medicine at the facility. They usually send us medicines quarterly but during COVID, deliveries often took more than 6 months to come. When patients come and don't find any medicines, they get frustrated." PR 11*

The participants supported that the lack of medications in public facilities was exacerbated by reduced income during COVID-19 lockdowns, including loss of jobs, reduced transportation of agricultural products to markets, and reduced commercial activities [27]. This reduced income for participants severely limited patients' ability to buy needed medications in the private pharmacies, ultimately impacting patients' health.

*"COVID-19 really made my life miserable. I did not know what was going to happen next; no money, no movements, no medicine. I was too weak. I was not myself and I was always dizzy. I can tell you l almost lost hope. PA 19*

*"It was a total mess, most of our colleagues lost their lives because they could not access their drugs and there was no one to think about them, everyone was just for him or herself, only the strongest survived. Some of us just survived by the mercies of God. We practically lived in constant fear, hopeless lives and with a lot of worries in every aspect of life." PA 16*

**2. Food insecurity.** The WHO recommends that patients with NCDs eat a balanced diet that is tailored to and help manage their disease [28]. Participants noted that disruptions in supply chains during COVID-19 lockdowns, led to food being significantly more limited and more expensive, especially fruits and vegetables. This food limitation was worsened by decreased patient income during lockdowns, as noted above.

*"There was no money during the COVID-19 lockdown. We did not have a variety of food and so we used to just eat whatever we came across." PA 5*

"*We used to get hungry and there was nothing to do, only to stare at each other and wait for whatever comes on our way. When children asked us for what to cook, we used to pretend as if we have not heard anything and provide whatever we could. If we managed to get some food, it would not enough.*" PA 8

"*There was no food supply, people were not working, no money and people were basically yearning for whatever they could just throw in their stomachs.*" PR 1

### 3. Barriers to delivery of clinical care

Patients and providers both highlighted challenges with NCD care that were exacerbated during the COVID-19 pandemic. For patients, difficulty with transportation to clinic was a significant barrier. Within the clinic, changes in clinical priorities, favoring COVID-19 care and lack of adequate personal protective equipment (PPE) meant that care for other diseases, such as NCDs, was compromised.

*3a. Transportation difficulties.* Transportation to and from clinic was a major barrier to care for the patients we interviewed, worsened by inadequate funds.

"*Remember most of us we are from poor societies with no money. In fact, after this interview, I am going to use my feet and walk home, which is about 10 kilometres from here.*" PA 8

Participants commented on the difficulty in moving from place to place during the COVID-19 lockdowns since the government severely limited people's movement. Motorcycle taxis "boda bodas," are the cheapest and most utilized mode of transportation in Uganda, but during lockdowns, they were affected by travel limitations and either didn't run or were more expensive than prior to COVID-19 pandemic.

"*Transport was restricted and most of the people who were able to come to the facility were those that came from nearer and those who were caught up by the lock down near this facility.*" PR 1

"*There is no way I could access the hospital, actually there was no transport, all movements were restricted, I had to just remain at my home without medicine.*" PA 13

"*Patients living with non-communicable diseases would not come to the hospital to seek treatment, they got so sick. Some of them ended up dying in their homes with no help in the villages and communities.*" PR 4

*3b. Clinic changes during COVID-19 lockdowns.* Participants encountered new clinic challenges that arose during COVID-19 lockdowns. Firstly, COVID-19 understandably became a clinical priority, but sadly at the expense of other services, like NCD care.

"*All the attention at the hospital was shifted from hypertension and other illnesses to COVID-19 strictly.*" PA 8

"*We were so concerned about COVID-19 that we gave little attention to other diseases. The quality of care was affected, and health education was no more.*" PR 10

Additionally, given a wide-spread lack of PPE in Uganda, there were legitimate concerns for contracting COVID-19 in healthcare facilities. Patients strategically avoided going to healthcare facilities to reduce their exposure to COVID-19 infection. Providers, in turn, limited interactions with patients to reduce their exposure.

"*There was also an element of stigma. Patients used to fear coming to the facility because they thought they would catch COVID-19.*" PR 5

"*We started fearing fellow human beings. As health workers, we would limit physical contact with patients in the health facility, and some of the vital measurements like the blood pressure, height, weight, and others were not done unless the patient was in a critical condition.*" PR 8

**4. Alternative forms of care.**   During the pandemic, the worsened access to medications and care led many patients to seek alternative forms of care, such as local herbs:

"*I had to look for alternative means so that I can stay alive and that was the local herbs, I used those local herbs through the whole COVID-19 lockdown until things got better and I was able to access hypertension drugs again.*" PA 8

"*For the hypertension patients, everyone is for him or herself, those who could manage, would come and others would buy medicine from the pharmacies near them. Then those who did not have money had to give up and use local herbs. Most of them missed doctor's consultations, health education and physical examination, they missed all that.*" PR 5

Additionally, patients sought care in private clinics that often did not have the expertise to care for their disease.

"*It was not comfortable getting treatment from those [private] clinic because they never used to test or examine us, instead they would just give us drugs and that was all.*" PA 9

## Recommendations for NCD care improvement

In discussing barriers to NCD care, participants highlighted how they felt NCD care could be improved, especially during lock-down situations. Recommendations from both patients and providers highlighted a need to make care more accessible and available closer to where patients lived. Participants also felt excited at the promise of integrating technology into NCD care, and especially mobile technology. Finally, participants often looked to innovations in HIV care as examples of providing effective longitudinal care.

**Improve the accessibility of care.**   Both the patients and providers we interviewed stated that care for chronic NCDs should be brought to lower-level clinics that are closer to where patients live. Participants supported that, before and after the pandemic, transportation was such a monumental barrier to care. They therefore reasoned that decentralizing care would make accessing care much more affordable, both in terms of time and money.

"*Hypertension drugs should be brought to health centres II and III so that those who cannot afford going to the main hospital also have a chance of accessing care without spending a lot on transportation. That will save [patients] money, time, and effort to travel such long distances.*" PA 3

As noted above, frequent medication stockouts compromise the Ugandan government's goal to provide basic NCD medications free-of-charge to patients. Patients and providers both stated that, to improve NCD care, medications should be more available and free-of-charge.

"*The government should have done a lot to help us, provide us with free medicines and bring them closer to health centres that are accessible and easy to reach*" PA 2

**Integrating technology.** During interviews, participants also supported the idea of using mHealth to help improve NCD care. Participants felt that mHealth could advance NCD care in several ways, especially care coordination (i.e., appointment scheduling), medical counseling, disease education, and adherence support. Participants highlighted that mHealth interventions would be especially helpful during lockdown periods when accessing healthcare facilities might be limited.

"*[Health workers] can also use telephone contacts to reach out to patients, ask them how they are feeling and get feedback as soon as possible. They can also advise patients, give health education, and provide reminders by phone in case movements are restricted.*" PA 5

"*Almost every person these days owns a cell phone, and these phones can help in passing on information, health education counselling, and reminders to people out there very cheaply and faster.*" PR 3

Providers felt that integrating mHealth into clinical care would help streamline care for patients with NCDs and help reduce the clinical burden on healthcare workers.

"*I think the use of technology in the provision on health services to patient with non-communicable diseases is a good idea and it will help in reducing the workload and reduce on the burden that are faced by patients. Technology also will help us both the client and the health workers do the right thing and improve on service delivery.*" PR 6

**Leveraging the care system developed for treating HIV.** During interviews, participants frequently referenced HIV care in Uganda as a example of how NCD care could be improved. Patients and providers noted that HIV patients had medications free-of-charge as well as community outreach to help with disease education and adherence support.

"*Patients living non-communicable diseases need constant and serious follow up just like other disease like HIV. We could try to engage them in community follow-up, sensitise them, and teach them the best way of taking their medicine very well.*" PR 4

Additionally, participants noted that HIV positive patients in Uganda have more social supports than patients with NCDs. This support includes organizations like The AIDS Support Organization (TASO), an HIV patient-support group that provides psychosocial and clinical services to supplement care provided at healthcare facilities [29].

"*There are some implementing partners who facilitate care like TASO and others. Most that is done with HIV patients, but I believe if the same can be done with people living with non-communicable diseases, it would really help a great deal.*" PR 2

## Discussion

In this qualitative study exploring the lived experience of NCD patients and healthcare providers during the COVID-19 pandemic, we found that NCD care delivery was severely disrupted during COVID-19 lockdowns in Uganda due to significant barriers to care, including an inability to access medication, food insecurity, lack of transportation, and disruptions in clinic workflows. These findings support prior work in Uganda that found reduced patient attendance to NCD clinics during COVID-19 lockdowns [30]. Additionally, our results are

consistent with multi-country studies on COVID-19 lockdowns that highlighted worsened patient barriers to care and severe disruptions in NCD care across SSA [10, 31].

The patients and providers we interviewed also had several key recommendations on how to improve NCD care: decentralize care, use novel technologies, and leverage the care system developed for treating HIV. We believe these approaches should be a blueprint for improving the day-to-day delivery of NCD care in southwestern Uganda and will also help mitigate disruptions in future societal lockdowns.

Barriers to NCD care herein emphasize the need to decentralize NCD care in SSA and strengthen supply chains [32]. Such decentralization would aim to ensure that drugs are available and affordable, and that care is found locally, near where patients live. In Uganda, fixed quantities of essential medications are shipped to facilities based on facility size. This system of drug delivery is ineffective with studies in Uganda showing that most facilities have frequent stockouts of both essential NCD medications and the majority lack the necessary diagnostic tools to care for NCD patients [17, 33]. Previous work has demonstrated that changing to a system where public clinics order and receive medicines directly from the central agency leads to dramatic reductions in stockout frequency and duration [34]. Additionally, projects like the Academic Model Providing Access to Healthcare in Kenya (AMPATH) have pooled fees from users in their catchment area to create pharmacy funds to successfully ensure access to essential medications, even during the COVID-19 pandemic [35]. Such creative approaches must be tried to increase the reliability of supply chains around essential medications in Uganda and throughout SSA.

Given how severely limited transport to clinic was during lockdowns, participants in this study highlighted how novel technology—especially mHealth—can significantly improve access to care when patient movement is limited. mHealth has shown significant promise in improving access to longitudinal NCD care in SSA, with numerous successes during the COVID-19 pandemic [36, 37]. Specifically, mHealth has shown promise in bolstering the health workforce, with Ebbs and colleagues showing effectiveness of combining in-person training with a mobile application to improve community health worker skills in northern Uganda. mHealth also supports self-care, with Schwartz and colleagues implementing a patient-facing mHealth application for heart failure patients in Uganda that dramatically improved both patient symptoms and exercise capacity [21]. While mHealth applications hold significant promise, implementing teams need to build trust with communities, especially in rural areas, to ensure that interventions are most effective [38].

In addition to mobile technology, integrating electronic health systems (EMRs) into NCD clinics in Uganda is imperative. EMRs allow providers to track individual patient clinical progress, but also to efficiently follow patient populations. In Uganda, EMRs have been shown to improve patient retention in HIV, and integrating an EMR into Kenyan health facilities improved hypertension and diabetes care through improved follow-up and standardization of care [39, 40]. Novel approaches to building EMRs in low-resource settings is critical, given issues with reliable power and internet connectivity. As an example, Feldacker and colleagues recently piloted a mobile EMR for community nurses in Malawi that provided decision-making support while improving data collection [41].

We do not need to start from scratch in starting to improve NCD care, as we have the blueprint from the global response to the HIV epidemic. There have been numerous calls to utilize the HIV infrastructure to extend longitudinal care to NCD patients [42]. One part of HIV infrastructure that participants in this study highlighted was the importance of social support through patient-support groups. Patient-support groups are voluntary organizations of patients and make patients active partners in managing their health [43]. Patient-support groups reliably increase patient retention, improve clinical outcomes and are incorporated

into WHO HIV guidelines [44]. During COVID-19 lockdowns, Ugandan HIV patients had drugs delivered to their homes through patient-support groups. Such patient-support groups have shown promising early results for NCD patients, though have yet to be widely adopted [45, 46]. We believe patient support groups would improve overall NCD care, and our research team is currently working on integrating them into clinics in southwestern Uganda.

A strength of this study is that we interviewed both patients and healthcare providers, getting a broad view of NCD care and the dire impact of COVID-19 and societal lockdowns. Additionally, while other studies have often focused on one NCD, we interviewed patients with several NCDs (hypertension, diabetes, and heart failure). While these diseases are unique, effective longitudinal care for all of them requires similar infrastructure and interventions that include functional medication supply chains, adequately trained healthcare professionals, and systems for patient tracking and follow-up [47]. A weakness of this study was that it involved patients only from southwestern Uganda who had access to the MRRH outpatient clinic. Additionally, results may have been affected by our roles as physicians from Uganda and the United States who have worked in southwestern Uganda for many years. However, we the authors have worked as physicians in rural areas across SSA and feel the themes in this study are far from unique to southwestern Uganda. NCD care across the sub-continent remains centralized and underfunded, contributing to significant morbidity and mortality that must be addressed [3, 48].

In conclusion, we found significant barriers to longitudinal NCD care in southwestern Uganda that were severely exacerbated during COVID-19 lockdowns. Future work should leverage mHealth and experiences caring for patients with HIV, with an eye towards de-centralizing NCD care, improving follow-up, and rectifying supply chain issues. Additionally, patient support groups could ensure that patients have social support when they need it most. Future research should aim to better understand rural versus urban differences in barriers to NCD care in southwestern Uganda, as well as leverage implementation science to evaluate the strategies recommended by the participants. Given the societal lockdown in Uganda for Ebola in late 2022 through early 2023, we must prepare for future lockdowns and ensure that NCD patients receive the care they need across the globe.

## Supporting information

**S1 Checklist. COREQ (COnsolidated criteria for REporting Qualitative research) checklist.**
(PDF)

**S1 File. Interview guides.** The interview guides used for participant interviews.
(PDF)

**S2 File. Master codebook.** The Master Codebook used to analyze the study's data.
(PDF)

## Acknowledgments

We would like to thank the health care providers at the Mbarara Regional Referral Hospital for their support of this work and their ongoing dedication to providing high quality patient care. We would also like to thank the Global Health Collaborative for ongoing administrative support for research in Mbarara, Uganda.

## Author Contributions

**Conceptualization:** Peter K. Olds, Paul S. Obwoya, Edwin Nuwagira, Jessica E. Haberer, Samson Okello.

**Data curation:** Peter K. Olds, Gabriel Nuwagaba.

**Formal analysis:** Peter K. Olds, Gabriel Nuwagaba.

**Funding acquisition:** Peter K. Olds.

**Methodology:** Peter K. Olds, Gabriel Nuwagaba, Jessica E. Haberer.

**Project administration:** Paul S. Obwoya, Edwin Nuwagira.

**Supervision:** Jessica E. Haberer, Samson Okello.

**Validation:** Jessica E. Haberer, Samson Okello.

**Writing – original draft:** Peter K. Olds.

**Writing – review & editing:** Paul S. Obwoya, Edwin Nuwagira, Jessica E. Haberer, Samson Okello.

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
