## [Decision Letter · Decision Letter 0]

9 Nov 2023

PONE-D-23-19925Patient-provider experiences with chronic non-communicable disease care during COVID-19 lockdowns in rural UgandaPLOS ONE

Dear Dr. Peter,

Thank you for submitting your manuscript to PLOS ONE. After careful consideration, we feel that it has merit but does not fully meet PLOS ONE’s publication criteria as it currently stands. Therefore, we invite you to submit a revised version of the manuscript that addresses the points raised during the review process.

Kind regards,

Desire Aime Nshimirimana, MBChB,Msc

Academic Editor

PLOS ONE

Journal Requirements:

"JEH reports consulting fees from Merck and stock ownership in Natera. All other authors declare none."

Additional Editor Comments:

Title: Kindly identify the topic as: A qualitative study

Abstract

The abstract is well written

Introduction

The introduction is too short. Add a research question/objective at the end of the introduction. Kindly follow Review1 comment on introduction

Methodology

1. Separate recruited participants: a. Patients and b. Providers: kindly explain in details how each group of participants was recruited. For provider participants, it seems you have used a snowballing method where a manager refers the interviewer to a specific provider participant among his/her workers. Kindly make it clear. For patient participants, it seems this was purposeful sampling, kindly make it clear how the sampling process was conducted for each group in a separated paragraph

2.Kindly add a paragraph of Inclusion & exclusion criteria for both groups, a. patient participants and b. provider participants

3. For interviews, we understand some participants were interviewed in Local language (Runyankore and Rukiga) and English. Clarify how you translated and translated back the questionnaires/results from local language to English after interviews

4. There should be a strong justification why you included a lab technician and a medical record officer. For a midwife, unless she/he is a health center manager or deals with chronic diseases in pregnancy, yet it is clear that pregnant women were excluded. A strong justification is needed. For example, a laboratory technician can know when the reagents are available but rarely know if chronic disease medicines are available or not. (For example, if the laboratory technician participates in Health center management meetings, he/she can get the right information). Kindly make it clear

I don’t comeback to “reflexivity”, reviewer1 has commented on this.

Findings

I am having a big concern that all participants (both providers and patients) can be recognized easily by reading this paper. Anyone from the Mbarara hospital who reads this paper can recognize most of the participants. How? In the interview, you only have one clinical medical records officer and a clinical laboratory technician. These participants are known to have participated in the study and may be recognized easily.

Reporting both (gender and age) expose patients to be recognized by the hospital managers in a small sample size. These characteristics expose them to their identity.

To avoid the risk of exposing participants, kindly identify their quotes using a code. For example; for providers, allocate PR and add a number (PR1: provider1, PR2,…) and patients (PA1, PA2, PA3: patient1, patient2,…..). You can decide to code any how you want. Then keep the legend of your codes in your records or dataset

The following are the reported quotes. Make sure to change for all quotes with codes.

“ Line 169: (female patient, age 86) and Line 174 (Male patient, 70), Line 183: clinical nurse, Line 205(female patient, 53), Line 210-211 (male patient, 66), Line 221 (male patient, 70), Line 226 (female patient, 56), Line 228 (Clinic midwife), Line 244 (Female patient, 58), Line 254 (Clinic midwife), Line 257 (Female patient, 63), Line 261 (Clinic medical records officer), Line 269 (Female patient, 58), Line 272 (Clinic nurse), Line 293 (Female patient, 58), Line 299 (clinic nurse), Line 305-306 (male patient, 47), Line 325 (Female patient, 38), Line 332 (Female patient, 40), Line 344 (Male patient, 70),Line 349 (Clinic nurse), Line 357 (Clinic nurse), Line 367 (Clinic medical records officer), Line 377 (Clinic nurse)”

You are doing a good synthesis in between the findings but you are not supporting your arguments with citations.

Kindly support your arguments with good academic support (i.e : line 198-201: this strong arguments need to be supported by a citation). Some paragraphs need a synthesis with good arguments supported by citations instead of a simple reporting.

Line 214-217: This good argument needs a citation support

Line 233: Kindly use only COVID-19 pandemic: not epidemic

Line 280-Line 285; Is this same clinical nurse? Clinical nurses are many and they are only reported as “clinical nurse”, therefore confusing whether is the same nurse or not. Reason why the coding is required.

Line 308: The heading “5. Recommendations for NCD care improvement”. The numbering of this heading is confusing. It has the numbering like “Themes”. This is number 5 making impression that it is “theme5” yet it is not. Kindly check and remove the confusion on numbering

Line 317-320: Kindly support these arguments with a citation

Discussion & Conclusions: The discussion & conclusions are well done

Kindly follow PLOS ONE formatting and revise English grammar, typos and errors

https://journals.plos.org/plosone/s/submission-guidelines

Reviewers' comments:

Reviewer's Responses to Questions

**Comments to the Author**

1. Is the manuscript technically sound, and do the data support the conclusions?

Reviewer #1: Yes

2. Has the statistical analysis been performed appropriately and rigorously? 

Reviewer #1: N/A

3. Have the authors made all data underlying the findings in their manuscript fully available?

Reviewer #1: Yes

4. Is the manuscript presented in an intelligible fashion and written in standard English?

Reviewer #1: Yes

5. Review Comments to the Author

Reviewer #1: PEER REVIEW OF MANUSCRIPT FOR PLOS ONE

TITLE: Patient-provider experiences with chronic non-communicable disease care during1

COVID-19 lockdowns in rural Uganda

ABSTRACT:

The abstract is concise and well written

INTRODUCTION

The introduction is very short, resulting an inadequate justification for the study. Many of the references cited in the discussion, could be brought into the introduction, as this was known already, including the impacts of lockdowns on NCDs in other countries, and the possibilities of mHealth. A funnel of existing evidence, starting with a worldwide perspective, and narrowing to LMICs and Africa, then to Uganda, would be helpful to set the scene. It is clear from the interview guide that the discussions about mHealth were the intention from the start, so bringing that literature into the introduction would be more appropriate.

METHODS

The choice of patients attending Mbarara Regional Referral Hospital and HC IV needs greater justification, as these are likely to be a specific group of patients requiring a higher level of care than those with uncomplicated NCDs who might be managed at a lower level of care. Was this a convenience sample, or was this a deliberate choice?

The selection of healthcare provider participants who were “purposively recruited in conjunction with the head of each health facility” needs further elaboration, in terms of the criteria for inclusion or exclusion beyond those stated in lines 95 to 98, as this could have unintentionally introduced a particular reflexivity bias. How a medical records officer and a laboratory technician based in a hospital, for example, would have particularly deep insight into NCD care during the lockdowns, is difficult to understand, compared to a community health worker for example.

The role and positions of the authors conducting a qualitative study, needs to be stated in the methods section, in terms of their positionality in relation to the participants and how this was mitigated, in terms of reflexivity. Then in the discussion section, the ways in which the authors who “have worked as physicians in rural areas across SSA” might have influenced the results, deserves a mention alongside any other limitations of the study.

RESULTS

These are well presented, with sufficient verbatim quotes to back up each statement.

DISCUSSION

The discussion is solid but not ground-breaking. What might make the whole manuscript more impactful would be to shift some of the references to the introduction, allowing space for a more critical discussion in relation to more recent literature in the topic. This should be followed, as mentioned above, by a reflection on the limitations of the study, as well as directions for future research. This goes beyond preparing for future lockdowns, to the implications arising from the study for routine NCD care outside of exceptional circumstances such as a pandemic.

REFERENCES

These need a lot more attention to detail in terms of completeness and formatting. There are too many shortcomings to mention each in detail, and this section should be checked for accuracy after revisions.

TYPO’S

Line 35: pre-existing

Line 110: NCD care

Line 309: “pain points” is a particular phrase that requires modification for an international audience

Line 398: strengthen (not strengthening)

6. PLOS authors have the option to publish the peer review history of their article (what does this mean?). If published, this will include your full peer review and any attached files.

Reviewer #1: **Yes: **Prof Steve Reid

---

## [Author Response · Author response to Decision Letter 0]

18 Nov 2023

Editor: 

RESPONSE: Thank you. I have updated the manuscript to meet the style requirements. 

"JEH reports consulting fees from Merck and stock ownership in Natera. All other authors declare none."

RESPONSE: The Competing Interest statement has been edited and added to the cover letter. 

 RESPONSE: Thank you. I have responded to this point in the cover letter. 

 RESPONSE: I have included a caption for the Supporting Information files.

 RESPONSE: Thank you. I have reviewed the reference list, and it is complete and correct. 

Additional Editor Comments:

Title: Kindly identify the topic as: A qualitative study

Abstract

The abstract is well written

Introduction

The introduction is too short. Add a research question/objective at the end of the introduction. Kindly follow Review1 comment on introduction

RESPONSE: Thank you for these comments. I have updated the title, and the introduction has been broadened to include additional background and a clearer research question/objective. 

Page 4-5: “We conducted a qualitative assessment of patients with hypertension, diabetes, and heart failure as well as the providers who care for them in southwestern Uganda, to evaluate how barriers to care changed during the COVID-19 lockdowns and understand perspectives on how to improve longitudinal NCD care, with a focus on mHealth integration.”

Methodology

1. Separate recruited participants: a. Patients and b. Providers: kindly explain in details how each group of participants was recruited. For provider participants, it seems you have used a snowballing method where a manager refers the interviewer to a specific provider participant among his/her workers. Kindly make it clear. For patient participants, it seems this was purposeful sampling, kindly make it clear how the sampling process was conducted for each group in a separated paragraph

RESPONSE: Thank you. I have separated the recruitment for patients and providers. For providers, we did not interview the head of the facilities, so we utilized purposeful sampling rather than snowball sampling. I have clarified this. 

Page 6: “Healthcare provider participants were, in turn, purposively recruited in conjunction with the head of each health facility. We did not interview the head of facility, so snowball sampling was not utilized.”

2. Kindly add a paragraph of Inclusion & exclusion criteria for both groups, a. patient participants and b. provider participants

 RESPONSE: I have added separate paragraphs for both sets of participants. 

3. For interviews, we understand some participants were interviewed in Local language (Runyankore and Rukiga) and English. Clarify how you translated and translated back the questionnaires/results from local language to English after interviews

RESPONSE: Thank you for highlighting this. I have clarified the translation and transcriptions process. 

4. There should be a strong justification why you included a lab technician and a medical record officer. For a midwife, unless she/he is a health center manager or deals with chronic diseases in pregnancy, yet it is clear that pregnant women were excluded. A strong justification is needed. For example, a laboratory technician can know when the reagents are available but rarely know if chronic disease medicines are available or not. (For example, if the laboratory technician participates in Health center management meetings, he/she can get the right information). Kindly make it clear

I don’t comeback to “reflexivity”, reviewer1 has commented on this.

RESPONSE: Thank you for raising this issue. I agree that it would have been ideal to have all nurses or physicians as interviewed providers. We utilized a convenience sample and ensured that providers interviewed had significant knowledge of NCD care at their facility. Given understaffing at Ugandan clinics, there is significant task shifting with providers workout beyond their defined scope of work. I have updated our methods section to specifiy this, and added more specificity in sampling and recruitment.

Page 5: “Staffing at health care facilities in Uganda is inadequate, with providers working multiple roles, often outside of their specialty.[22] Thus, despite defined roles, providers at health centers often have significant knowledge of NCD care, either through task shifting and providing direct care—for mid wives— or by managing other aspects of NCD care and/or participating in health center management meetings—for medical records officers and laboratory technicians.”

Page 6: “This was a convenience sample and, since we did not interview heads of facility, snowball sampling was not utilized. We aimed for a balanced distribution of participants from rural and urban clinics.”

Findings

I am having a big concern that all participants (both providers and patients) can be recognized easily by reading this paper. Anyone from the Mbarara hospital who reads this paper can recognize most of the participants. How? In the interview, you only have one clinical medical records officer and a clinical laboratory technician. These participants are known to have participated in the study and may be recognized easily. Reporting both (gender and age) expose patients to be recognized by the hospital managers in a small sample size. These characteristics expose them to their identity.

To avoid the risk of exposing participants, kindly identify their quotes using a code. For example; for providers, allocate PR and add a number (PR1: provider1, PR2,…) and patients (PA1, PA2, PA3: patient1, patient2,…..). You can decide to code any how you want. Then keep the legend of your codes in your records or dataset

The following are the reported quotes. Make sure to change for all quotes with codes.

“ Line 169: (female patient, age 86) and Line 174 (Male patient, 70), Line 183: clinical nurse, Line 205(female patient, 53), Line 210-211 (male patient, 66), Line 221 (male patient, 70), Line 226 (female patient, 56), Line 228 (Clinic midwife), Line 244 (Female patient, 58), Line 254 (Clinic midwife), Line 257 (Female patient, 63), Line 261 (Clinic medical records officer), Line 269 (Female patient, 58), Line 272 (Clinic nurse), Line 293 (Female patient, 58), Line 299 (clinic nurse), Line 305-306 (male patient, 47), Line 325 (Female patient, 38), Line 332 (Female patient, 40), Line 344 (Male patient, 70),Line 349 (Clinic nurse), Line 357 (Clinic nurse), Line 367 (Clinic medical records officer), Line 377 (Clinic nurse)”

Line 280-Line 285; Is this same clinical nurse? Clinical nurses are many and they are only reported as “clinical nurse”, therefore confusing whether is the same nurse or not. Reason why the coding is required.

RESPONSE: Thank you for this thoughtful comment, and we agree that the participants can be recognized too easily. I have used the coding system you recommend and updated each quote accordingly. 

You are doing a good synthesis in between the findings but you are not supporting your arguments with citations.

Kindly support your arguments with good academic support (i.e : line 198-201: this strong arguments need to be supported by a citation). Some paragraphs need a synthesis with good arguments supported by citations instead of a simple reporting.

Line 214-217: This good argument needs a citation support

Line 317-320: Kindly support these arguments with a citation

 RESPONSE: We agree that these sentences needed further clarification. 

Line 198-201 (now 317-321): “The participants supported that the lack of medications in public facilities was exacerbated by reduced income during COVID-19 lockdowns, including loss of jobs, reduced transportation of agricultural products to markets, and reduced commercial activities.[25]”

Line 214-217 (now 337-341): “The WHO recommends that patients with NCDs eat a balanced diet that is tailored to and help manage their disease [26]. Participants noted that disruptions in supply chains during COVID-19 lockdowns, led to food being significantly more limited and more expensive, especially fruits and vegetables.”

Line 317-320 (now 462-466): “Both the patients and providers we interviewed stated that care for chronic NCDs should be brought to lower-level clinics that are closer to where patients live. Participants supported that, before and after the pandemic, transportation was such a monumental barrier to care. They therefore reasoned that decentralizing care would make accessing care much more affordable, both in terms of time and money. 

Line 233: Kindly use only COVID-19 pandemic: not epidemic

 RESPONSE: This has been changed. 

Line 308: The heading “5. Recommendations for NCD care improvement”. The numbering of this heading is confusing. It has the numbering like “Themes”. This is number 5 making impression that it is “theme5” yet it is not. Kindly check and remove the confusion on numbering

RESPONSE: Thank you for identifying this. We agree that this is confusing and have removed numbering from the final category “Recommendations for NCD care improvement.”

Discussion & Conclusions: The discussion & conclusions are well done

Reviewer 1:

ABSTRACT:

The abstract is concise and well written

INTRODUCTION

The introduction is very short, resulting an inadequate justification for the study. Many of the references cited in the discussion, could be brought into the introduction, as this was known already, including the impacts of lockdowns on NCDs in other countries, and the possibilities of mHealth. A funnel of existing evidence, starting with a worldwide perspective, and narrowing to LMICs and Africa, then to Uganda, would be helpful to set the scene. It is clear from the interview guide that the discussions about mHealth were the intention from the start, so bringing that literature into the introduction would be more appropriate.

RESPONSE: Thank you for such a careful reading. The introduction has been expanded to add additional context. 

METHODS

The choice of patients attending Mbarara Regional Referral Hospital and HC IV needs greater justification, as these are likely to be a specific group of patients requiring a higher level of care than those with uncomplicated NCDs who might be managed at a lower level of care. Was this a convenience sample, or was this a deliberate choice?

RESPONSE: Thank you for highlighting this. NCD care in Uganda is quite centralized, so recruitment at MRRH was both a convenience sample and deliberately choice, as the patients are relatively representative. I have clarified these points as noted below: 

Page 5: “Due to the limited reliability in access to NCD care and medications in the region, longitudinal NCD care remains centralized, with significant numbers of patients seeking care at the MRRH outpatient clinics where resources are more often available.” 

Page 6: “This was a convenience sample, though given the centralized nature of NCD care in southwestern Uganda we describe above, the MRRH outpatient clinics treat a relatively representative sample of patients in the region.”

The selection of healthcare provider participants who were “purposively recruited in conjunction with the head of each health facility” needs further elaboration, in terms of the criteria for inclusion or exclusion beyond those stated in lines 95 to 98, as this could have unintentionally introduced a particular reflexivity bias. How a medical records officer and a laboratory technician based in a hospital, for example, would have particularly deep insight into NCD care during the lockdowns, is difficult to understand, compared to a community health worker for example.

RESPONSE: Thank you for raising this issue. I agree that it would have been ideal to have all nurses or physicians as interviewed providers. We utilized a convenience sample and ensured that providers interviewed had significant knowledge of NCD care at their facility. Given understaffing at Ugandan clinics, there is significant task shifting with providers workout beyond their defined scope of work. I have updated our methods section to specifiy this, and added more specificity in sampling and recruitment.

Page 5: “Staffing at health care facilities in Uganda is inadequate, with providers working multiple roles, often outside of their specialty.[22] Thus, despite defined roles, providers at health centers often have significant knowledge of NCD care, either through task shifting and providing direct care—for mid wives—or by participating in health center management meetings—for medical records officers and laboratory technicians.”

Page 6: “This was a convenience sample and, since we did not interview heads of facility, snowball sampling was not utilized. We aimed for a balanced distribution of participants from rural and urban clinics.”

The role and positions of the authors conducting a qualitative study, needs to be stated in the methods section, in terms of their positionality in relation to the participants and how this was mitigated, in terms of reflexivity. Then in the discussion section, the ways in which the authors who “have worked as physicians in rural areas across SSA” might have influenced the results, deserves a mention alongside any other limitations of the study.

 RESPONSE: I have added the role and position of the authors, both in the methods section, and as a limitation in the discussion. 

 Pages 7-8: “The interviewer, GN, is bachelor trained, trilingual, and a non-clinician. The interview guide was developed by author PKO with support from PSO, EN, JEH, and SO. Authors PSO, EN, and SO are Ugandan internal medicine physicians who work at MRRH, with intimate knowledge of NCD care in the region. PKO is an American internal medicine physician who has conducted research and provided clinical care in Uganda, Madagascar, Malawi, and Rwanda since 2011. JEH is American internal medicine physicians who has conducted research in Uganda, Kenya, and South Africa since 2008.”

 Page 23: “Additionally, results may have been affected by our roles as physicians from Uganda and the United States who have worked in southwestern Uganda for many years.”

RESULTS

These are well presented, with sufficient verbatim quotes to back up each statement.

DISCUSSION

The discussion is solid but not ground-breaking. What might make the whole manuscript more impactful would be to shift some of the references to the introduction, allowing space for a more critical discussion in relation to more recent literature in the topic. This should be followed, as mentioned above, by a reflection on the limitations of the study, as well as directions for future research. This goes beyond preparing for future lockdowns, to the implications arising from the study for routine NCD care outside of exceptional circumstances such as a pandemic.

RESPONSE: Thank you for such a critical read of the manuscript. As noted above, I have updated the introduction. Additionally, the discussion has been re-worked to highlight how these results can be applied to day-to-day NCD care and incorporating more recent literature. 

REFERENCES

These need a lot more attention to detail in terms of completeness and formatting. There are too many shortcomings to mention each in detail, and this section should be checked for accuracy after revisions.

RESPONSE: Thank you for this comment. The issue with the citation manager program has been addressed, and I believe that the citations are now properly formatted. 

TYPO’S

Line 35: pre-existing

Line 110: NCD care

Line 309: “pain points” is a particular phrase that requires modification for an international audience

Line 398: strengthen (not strengthening)

RESPONSE: Thank you for such a thorough review. I have corrected these typos, and “pain points” has been changed to “barriers.”

---

## [Editor Report · Decision Letter 1]

27 Nov 2023

Patient-provider experiences with chronic non-communicable disease care during COVID-19 lockdowns in rural Uganda: a qualitative analysis

PONE-D-23-19925R1

Dear Dr. Olds,

We’re pleased to inform you that your manuscript has been judged scientifically suitable for publication and will be formally accepted for publication once it meets all outstanding technical requirements.

Kind regards,

Desire Aime Nshimirimana, MBChB,Msc

Academic Editor

PLOS ONE

---

## [Editor Report · Acceptance letter]

5 Dec 2023

PONE-D-23-19925R1 

Patient-provider experiences with chronic non-communicable disease care during COVID-19 lockdowns in rural Uganda: a qualitative analysis 

Dear Dr. Olds:

I'm pleased to inform you that your manuscript has been deemed suitable for publication in PLOS ONE. Congratulations! Your manuscript is now with our production department. 

Kind regards, 

on behalf of

Dr Desire Aime Nshimirimana 

Academic Editor

PLOS ONE